# Feeding Habits of *Sarda chiliensis chiliensis* (Cuvier, 1832) in Northern Chile and Southern Perú

**DOI:** 10.3390/ani12070930

**Published:** 2022-04-05

**Authors:** Renzo Pepe-Victoriano, Héctor Aravena-Ambrosetti, Jordan I. Huanacuni, Felipe Méndez-Abarca, Karina Godoy, Nathalia Álvarez

**Affiliations:** 1Facultad de Recursos Naturales y Renovables, Universidad Arturo Prat, Avenida Santa María 2998, Arica 1031597, Chile; hector.aravena.ambrosetti@gmail.com (H.A.-A.); jhuanacunip@unjbg.edu.pe (J.I.H.); f.mendez.abarca@gmail.com (F.M.-A.); karinagodoycondori@gmail.com (K.G.); nathalia.montserrat@gmail.com (N.Á.); 2Programa de Doctorado en Acuicultura Sostenible y Ecosistemas Marinos, Instituto Universitario ECOAQUA, Universidad de Las Palmas de Gran Canaria, Crta. Taliarte s/n, 35214 Telde, Spain; 3Programa de Magíster en Acuicultura Mención, Cultivo de Recursos Hidrobiológicos y Mención Acuaponia, Facultad de Recursos Naturales y Renovables, Universidad Arturo Prat, Avenida Santa María 2998, Arica 1031597, Chile; 4Escuela Profesional de Ingeniería Pesquera, Facultad de Ciencias Agropecuarias, Universidad Nacional Jorge Basadre Grohmann, Av. Cusco s/n, Tacna 23000, Peru

**Keywords:** stomach content, South Pacific Bonito, feeding dynamics and strategy, composition and trophic relationships

## Abstract

**Simple Summary:**

Knowing the biology of fish, preferably for this study of trophic relationships, is fundamental to advance in the culture technologies of this species, since knowing what they feed on will allow us to generate a species-specific food for the species and thus be able to obtain a fast and efficient growth. The objective of this research was to analyze the feeding habits of South Pacific bonito by sex and season between summer and autumn in three locations in the South Pacific, using trophic biology techniques. A total of 1404 stomachs were analyzed, corresponding to 420 specimens from the Pozo de Lisas area, 540 specimens from the La Capilla area and 444 specimens from the Chanavayita area. Dynamic indexes of feeding analysis and trophic composition of the diet were used. Under the analysis carried out in this research, we can conclude that fish feed more during autumn; in addition, the prey hierarchy of this species was *Pleuroncodes monodon* and *Engraulis ringes*. That said, and in view of the results, it would be a generalist species.

**Abstract:**

The trophic relationships of pelagic fishes with migratory traits such as the South Pacific Bonito, *Sarda chiliensis chiliensis* (Cuvier, 1832), have not been studied in depth in the past. The objective of the present research was to analyze the feeding habits of South Pacific Bonito by sex and to analyze a comparison between summer and autumn months in three different areas of the eastern South Pacific by applying different techniques used in trophic biology. Between December 2013 and June 2014, specimens were captured in the areas of Pozo de Lisas (Ilo, Peru), La Capilla (Arica, Chile) and Chanavayita (Iquique, Chile). The feeding dynamics and trophic composition of the diet were analyzed, as well as the feeding strategy and trophic relationships. A total of 1404 specimens were analyzed, of which 654 had stomach contents. Seven prey items were identified: (a) fish remains; (b) squid jaw remains; (c) squid gladius remains; (d) caudal fin remains; (e) *Engraulis rigens*; (f) *Pleuroncodes monodon* and (g) N/A (not determined). The Pozo de Lisas and La Capilla areas showed homogeneity in their prey items, while the Chanavayita area showed more diversity. Regarding the importance of prey items in the diet of *S. chiliensis* in the three localities, it would be correct to state that it is a generalist species.

## 1. Introduction

The most appropriate means to study a species’ diet is through the analysis of the prey found in their stomachs [1]. Knowledge of the feeding habits of a species is a basic aspect in order to be acquainted with its biology [2] since they depend on anatomical, physiological and ethological adaptations. Knowing about a species’ feeding habits also represents the most certain path to know about their trophic relationships [3]. 

The category fish has a wider trophic niche than other vertebrates, and a high overlap of diet is observed, together with a low overlap of habitat. Both patterns, generalizable in fishes, condense to some extent the strategies and conflicts of the ichthyofauna in different environments [4]. 

Trophic ecology is necessary to understand the biology and ecology of organisms, where food is one of the most significant and influential factors [5]. Said understanding allows us to obtain information regarding the role a species plays in a determined ecosystem [6], through the knowledge of the consumed species, their preferences and the effect of their dynamics on the community [7]. The study of stomach content is a common way to investigate the food chains of a biological marine community [8].

In Chile, the first research on fish feeding habits was conducted on a species important to fisheries [9,10,11,12,13,14,15].

The trophic relationships of pelagic fishes with migratory characteristics, such as the South Pacific Bonito, have not been studied in detail, and some authors have analyzed only some aspects of the trophic dynamics [1,2]. Many authors only described quantitatively the diet of pelagic fishes and did not analyze the trophic variations among them and among different localities where the species were found. [16]. We consider the latter to be one of the most important intrinsic factors, since they regulate and affect growth and reproduction, as well as the development of their life cycle; a process that occurs at the expense of energy received from the outside [17,18]. 

The importance for the farming of this species lies in the instability of its fishery. Between 2010 and 2015, there was a notable decrease in its catches, of less than 30 tons. Currently, the South Pacific Bonito fishery in northern Chile has reached 131 tons [19] and in Peru, it has reached only 92 tons [20].

The South Pacific Bonito is a neritic epipelagic species [21], which inhabits the most temperate and saline surface layer of sub-Antarctic waters [22]. Its characteristics include: fusiform body, dark blue in color with a metallic sheen, five to nine dark dorsal to ventral obliquely oriented stripes, weight up to 5.5 kg, and a length up to 1 m. It is notable for its huge mouth and conical teeth, as well for its large, round eyes [23]; their diet includes other fish such as anchoveta (*Engraulis ringens*), Spanish sardine (*Sardinops sagax*), chub mackerel (*Scomber japonicus*), Chilean jack mackerel (*Trachurus murphyi*) and Atlantic saury (*Scomberesox saurus*) [24]. This is one example of why it is important to gather knowledge of the biology of *S. chiliensis chiliensis*, not only for species of economic value, as reported by Pepe-Victoriano et al. [25,26], but also for all those species with which they are ecologically related, since an alteration in their dynamics could directly or indirectly affect the survival of any species.

Within this frame, the significance of determining the feeding habits based on the background of stomach content studies is made clear. In this sense, our research sought to analyze the feeding habits of South Pacific Bonito *Sarda chiliensis chiliensis* by sex and comparison between summer and autumn months, in three areas of the eastern South Pacific through different techniques in trophic biology.

## 2. Materials and Methods

### 2.1. Capture and Study Area 

Between December 2013 and June 2014, individuals of *Sarda chiliensis chiliensis* (Figure 1) were collected by artisanal fishermen using purse seine off the coasts of (a) Pozo de Lisas, located seven kilometers south of the city of Ilo, Peru, (b) Playa La Capilla, located ten kilometers south of the city of Arica, Chile and (c) Playa Chanavayita, located sixty kilometers south of Iquique, Chile (Figure 2). Specimens were taken to the laboratory where they were identified based on meristic and morphometric characters proposed by Chirichigno [27].

### 2.2. Stomach Extraction

Each individual was dissected in order to extract their stomachs and then sexed. They were also searched for any food remains in the mouth and invagination in the stomach, that is, whether the stomach was empty, elongated and the walls were thin. 

The extraction of stomachs was performed following the methodology suggested by Amezaga [3] and Morte and Sanz [28], where the abdominal cavity is opened through a longitudinal incision in the ventral area, and then the intestine is cut at the pylorus level. Finally, the stomach is removed by performing a cut at the esophagus level.

Once each stomach was removed, it was weighed and immediately fixed in a 10% formalin solution neutralized with sodium borate at pH 7 and then preserved in 70% alcohol. Once opened, the prey were identified, counted and weighed.

A total of 1404 stomachs were analyzed, corresponding to 420 specimens from the Pozo de Lisas area, 540 specimens from the La Capilla area and 444 specimens from the Chanavayita area.

### 2.3. Treatment of Stomach Contents

The contents were laid in Petri dishes with seawater and were analyzed under a stereoscopic microscope (model XTL-2310). All prey were divided into major taxonomic groups, and the number and weight in grams of prey was recorded.

Individuals of each identified taxon were counted. In general, the units could be quantified through the precise identification of the remains. To avoid any errors in highly digested prey, the criterion used was the identification of structures that seemed to always be present in the stomachs, such as: calcified hard remains, eyes, exoskeletons, bones, otoliths, spines and skin remains. All this was in relation to standard samples and other stomach contents with a lower degree of digestion, considering, for otoliths, the limitations linked to their use [29,30].

The “not determined” section (N/D) includes all other remains of animal origin (except for squid and fish) that, due to their state of digestion, could not be identified within the established groups. 

After identification, a collection of reference samples of the prey was elaborated. This collection was composed of both whole specimens and fragments, fish otoliths, cephalopod beaks and feathers, hard parts in general, etc.

### 2.4. Food Dynamics Analysis

(a)Vacuity Index (VI)

To estimate the proportion of empty stomachs within the sample, the Vacuity Index was calculated [28,31,32,33,34]:(1)$$I_{v}=\left(\frac{E_{v}\;}{\;E_{t}}\right)\times 100$$
where:

*E_v_*: number of empty stomachs;

*E_t_*: total number of stomachs analyzed.

(b)Weight of stomach contents (*W_sc_*)

For each stomach, the weight of stomach contents was obtained through the difference between the weight of a full and empty stomach [33].
(2)$$W_{sc}=W_{t}-W_{p}$$
where:

*W_t_*: weight in grams of full stomach; 

*W_p_*: weight in grams of empty stomach.

Both weights were measured in stomachs preserved in 70% alcohol with a precision balance (±0.001 g).

(c)Mean Fullness Index (*IR*)

Since changes in stomach contents indicate differences in feeding intensity over time, the rate of feeding behavior was determined using the Mean Fullness Index, according to the methodology proposed by Hyslop [35], Collette and Nauen [21] and Cabral et al. [36].
(3)$$IR\;\left(I_{L1}\right)\left(\%\right)=\left(\frac{W_{sc}\;}{\;W_{E}}\right)\times 100$$
where:

*W_ce_*: weight in grams of stomach content; 

*W_E_*: weight in grams of eviscerated fish. 

(d)Average weight of the prey per stomach (*W_p_*)

Aguirre [34], proposed to estimate the average weight of prey per stomach (*W_p_*) as follows:(4)$$W_{p}=\left(\frac{W_{c}\;}{\;N_{p}}\right)$$
where:

*W_c_*: weight in grams of stomach contents.

*N_p_*: number of prey per stomach.

### 2.5. Study of the Trophic Composition of the Diet

Once prey items were identified, a quantitative analysis of food components was performed with the follow methods:(a)Numerical composition (*NC*%)

It expresses the proportion of each type of prey in relation to the total number of prey items found in the whole sample of analyzed stomachs [3,35]. This result allowed us to calculate the average number of individuals per food category and per stomach.
(5)$$NC\left(\%\right)=\left(\frac{n_{i}}{\;N_{p}}\right)\times 100$$
where:

*n_i_*: total number of representatives of prey;

*N_p_*: total number of prey consumed.

(b)Frequency of Occurrence (*F*%)

(6)$$F\left(\%\right)=\left(\frac{E_{i}\;}{\;E_{T}}\right)\times 100$$
where:

*E_i_*: number of stomachs with prey;

*E_T_*: total number of stomachs. 

(c)Gravimetric method (*G*%)

The results are expressed as a percentage of the weight of each food category versus the total weight of all stomach contents [28].
(7)$$G\left(\%\right)=\left(\frac{W_{i}}{\;W_{T}}\right)\times 100$$
where:

*W_i_*: prey weight;

*W_T_*: total weight of prey.

(d)Relative Importance Index (*RII*)

It is calculated by the sum of the percentages in number and volume and later multiplying the result by the percentage of frequency of occurrence for each type of food [37].
(8)RII=(CN %+G %)×F %
where: 

*CN*%: numerical composition;

*G*%: gravimetric method;

*F*%: frequency of occurrence.

(e)Geometric Importance Index (*GII*)

Applied in order to compare the results obtained in the different indexes and as a tool to rank prey that make up the diet of each species in order of importance [38].
(9)$$GII_{i}=\frac{{\left(\sum^{}Vi\right)}_{j}}{\sqrt{n}}$$
where:
*V_i_*: represents the value of each measurement of the relative quantity *i* of prey *i*, i.e., the value of each applied index (numerical, frequency and gravimetric).*n*: number of indexes applied.

### 2.6. Evaluation of Feeding Strategy

Among the customary methods to measure feeding strategy are that of Costello [39], best known for its great simplicity. It is based on the relative position occupied by different prey on a coordinate axis in which the abscissae correspond to the frequency of occurrence and the ordinates to the abundance represented by the gravimetric index.

### 2.7. Study of Trophic Relationships

(a)Shannon’s Diversity Index (*H*′)

Vicent and Aparici [32] and Aguirre [33] propose to determine the diet breadth of each species and size cluster through the Shannon’s Diversity Index [40].

That is, by using the numerical importance values of prey: (10)$$H^{\prime }=-\overset{n}{\underset{i=1}{\sum }}\;\left(p_{i}\;ln\;ln\;p_{i}\right)\;$$
where:
*i*: type of trophic resource;*p_i_*: is the number of species i, expressed as the proportion of the sum of *Pj* of prey species;N: number of prey or trophic resources. 

Values vary between 0 and 6, values <3 indicate a poorly diverse diet and >3 indicate a diverse diet.

(b)Homogeneity Index (*J*′)

The homogeneity of the diet was calculated by standardizing *H*′ to a scale between 0 and 1. The Shannon index of homogeneity was applied:(11)$$J^{\prime }=H^{\prime }\;/\;H_{max}$$
where:

*H_max_*: corresponds to the natural logarithm of the number of trophic components per sample.

### 2.8. Statistical Analysis

For the study of feeding dynamics and diet composition, several analysis techniques were implemented. Parametric tests such as ANOVA were applied to the data that conformed to a normal distribution (Kolmogorov–Smirnov test). Those that presented homogeneous variances (Levene test) were: weight of stomach contents (WCe), Mean Fullness Index (IR), and average weight of prey (Wp). For the latter, the identification of significant differences according to the different sampling sectors and sex was aimed.

Non-parametric techniques (X2 test and contingency tables) were also implemented in those cases in which the data did not comply with the previous premise, such as the Vacuity Index (VI). 

Prominent differences in the average number of prey (Np), and the Relative Importance Index of prey (RI%) with respect to variables such as different study locations and sex were evaluated by multiple regression analysis.

## 3. Results

Fish captured in the Chanavayita area had a greater average standard length than in the La Capilla and Pozo de Lisas areas, with a difference of 9 and 6 cm, respectively. The sex ratio was higher for males in the three capture areas (Table 1).

Prey were identified to the lowest taxonomic level possible. Cases in which it was not possible to determine the exact species of prey corresponded to highly digested and chopped samples, making it a challenge to determine their category.

Out of the 1404 analyzed stomachs, only 145 from Pozo de Lisas, 132 from La Capilla and 129 from Chanavayita showed stomach contents. Six prey items were identified from the stomach contents of the specimens: (a) fish skeleton remains; (b) cephalopod remains —identified using the morphology of squid jaws; (c) squid gladius (pen) remains; (d) *Pleuroncodes monodon*; (e) caudal fin remains; (f) *Engraulis rigens*; N/D (not determined). Similar to *Sarda chiliensis lineolata*, *Sarda chiliensis chiliensis* would play the same role in the trophic webs in its respective hemisphere, since a greater component of fish remains, represented by the vertebrae of anchovetas, was found (*Engraulis ringens*).

### 3.1. Analysis of Feeding Dynamics

#### 3.1.1. Vacuity Index (VI) and Mean Fullness Index (IR)

The results from this index enlightened the importance of empty stomachs within the sample. Analyzing the sample in the three studied areas, it was found that the highest number of empty stomachs occurred in the summer months. The highest Vacuity Index percentage was in the Chanavayita area with 59.09%, followed by La capilla with 56.92% and then Pozo de Lisas with 56.36% (Figure 3). In autumn, the Vacuity Index was lower in all three areas, 46.67% (Pozo de Lisas), 45.88% (Chanavayita) and 41.79% (La Capilla) (Figure 3). Although the Vacuity Indexes were different among the zones, there was no statistically significant difference. 

The Vacuity Index with respect to sex demonstrated that males had the highest vacuity percentage in La Capilla (50.67%), Pozo de Lisas (50.59%), and Chanavayita (47.14%) (Figure 3). Conversely, female individuals showed the highest percentage in Chanavayita (54.24%), followed by Pozo de Lisas (50.00%) and La Capilla (47.37%) (Figure 3), with no significant differences. 

Figure 4 demonstrates the fluctuations in the different study areas of Vacuity and Mean Fullness Indexes. 

#### 3.1.2. Weight of Stomach Contents (Wce)

Figure 4 shows the trends by weight of stomach content and Mean Fullness Index due to variations in seasonality and sex. There were no significant differences between the different seasons and between males and females.

#### 3.1.3. Average Weight of Prey per Stomach (Wp)

Figure 5 shows that the largest prey are consumed in summer. It can also be noted that males have the largest prey in their stomachs. Despite these inequalities, there are no significant differences.

Comparing in detail the weight of stomach contents and the Mean Fullness Index in the three study areas, we observed that there were significant differences in the amount of food present in the stomachs between summer and autumn (W_ce_: F_2.74_ = 3.62; *p* < 0.05; IR: F_2.74_ = 2.79; *p* < 0.05). Summer was the season with the lowest stomach content. The season with the highest consumption was autumn.

No significant differences were found between the sexes for the three areas (Wce: F_1.93_ = 0.68; *p* = 0.39; IR: F_1.93_ = 2.18; *p* = 0.18); however, it can be observed that the tendency is for females to have lower contents than males, as well in terms of their Vacuity Index.

### 3.2. Study of the Trophic Composition of the Diet

In order to establish the dietary composition of individuals from different locations and the significance of different prey, the Numerical Composition Index (N%), Frequency of Occurrence Index (F%), and Gravimetric Index (G%) were calculated, as shown in Table 2, Table 3 and Table 4. To complement and to be able to observe a hierarchy of the prey, the Geometric Importance Index (GII) proposed by Assis [38] was applied, yet another combined method that has a graphic representation.

#### 3.2.1. Pozo de Lisas Study Area

Geometric Importance Index (GII)

In order to rank and classify the prey from all three study areas, we have graphically condensed the information, using the Geometric Importance Index suggested by Assis [38] (Figure 6). 

#### 3.2.2. La Capilla Study Area

Geometric Importance Index (GII)

In order to hierarchize and classify prey that are part of the diet of La Capilla specimens, we have condensed the information in the following graph (Figure 7).

#### 3.2.3. Chanavayita Study Area

Frequency of Occurrence (F%)Geometric Importance Index (GII)

In order to rank and classify prey from the La Capilla area, we have condensed the information in Figure 8.

No significant differences were found in the number of prey items for the different indexes in the study areas with respect to sex and summer and autumn seasons.

### 3.3. Evaluation of the Food Strategy

Figure 9 shows the importance of each prey item in the different study areas, which determines whether they are dominant or rare, and their feeding strategy in terms of whether they are specialists or generalists. Regarding the importance of prey in the diet of *Sarda chiliensis chiliensis* in the Pozo de Lisas area, *Pleuroncodes monodon* was the dominant prey, and in La Capilla, *Engraulis ringens*, and in Chanavayita, *P. monodon* and *E. ringens.*

### 3.4. Study of Trophic Relationships

Shannon’s Diversity Index (H’) and homogeneity

Table 5 shows the overall values of diversity and homogeneity for the studied zones.

According to the results of Shannon’s Diversity Index (H′), Chanavayita was the area with the greatest food diversity for *Sarda chiliensis chiliensis*, as the studied specimens presented the greatest niche amplitude with respect to the other locations of sampling, followed by Pozo de Lisas and La Capilla, in that order.

Regarding homogeneity (J′), the study zones of Pozo de Lisas and La Capilla presented a value close to 1, which shows that the specimens in these areas present a greater diet homogeneity.

## 4. Discussion

The average standard lengths of South Pacific Bonito specimens between December 2013 and June 2014 were 41.2, 36.4 and 45.4 cm in the areas of Pozo de Lisas, La Capilla and Chanavayita, respectively. These measures could correspond to young specimens, since there exist reports of maximum length sizes of at least 79 cm in Bonitos from the southern hemisphere [25]. These small sizes during the reproductive period could coincide with a decrease in average size (length and weight) as a result of the decline in South Pacific Bonito populations due to overfishing. As reported by Pauly et al. [41], a decline in Bonito has been observed since 1987 for Peru and a similar decline in Chile [22], which has now been corroborated by SERNAPESCA [42], in Chile and by PRODUCE in Peru [20]. In this sense, the reduction in the average size of Bonito is directly related to the decline of anchoveta, its favorite prey according to Pauly et al. [41].

### 4.1. Analysis of Feeding Dynamics

In the analysis of feeding dynamics, some prey are rapidly digested and are therefore harder to detect. Moreover, prey such as crustaceans that have chitinous exoskeletons, fish vertebrae, otoliths and others remain identifiable for long periods of time. For this reason, the rigorous analysis of quantitative data has not yet been fully resolved. However, successful sampling selection and statistical methods can better illustrate the diet of a species, as hard prey are often overestimated over soft prey, which are likely to be underestimated [43].

Stoner [44] and Gibson [45] state that fish feeding dynamics are mainly governed by external factors such as temperature, light intensity and food abundance, as well by internal factors such as hunger and to some degree reproduction; many species do not feed during the spawning season.

Out of the 1404 extracted specimens, a high percentage had empty stomachs (Vacuity Index of 71.08%). This could be explained by the proximity of these Bonitos to the coast, as they might not have had access to preferred prey. These prey correspond to anchoveta (*Engraulis ringens*), Spanish sardine (*Sardinops sagax*), chub mackerel (*Scomber japonicus*), Chilean jack mackerel (*Trachurus murphyi*) and Atlantic saury (*Scomberesox saurus*) as described by Medina et al. [24]. Pauly et al. [41] established that the population dynamics of anchoveta determine the degree of consumption of South Pacific Bonito as a highly specialized predator of anchoveta; with data also established by other authors for highly specialized fish of the *Carangidae* family [46]. In turn, these authors [41,46] analyzed the data provided by various studies of the stomach contents of *Sarda chiliensis chiliensis*, among which are those of Chirinos [47], Ancieta [48] and Canal [49], who reported for the waters of Callao a Vacuity Index of 77%, 49.4% and 42.4%, respectively. Meanwhile, Mayo [50] reported 46.6% and Pauly et al. [41] reported 51.4% for specimens obtained along the Peruvian coast. In other words, there is a reported gradient of the Vacuity Index from 42.4% to 77%, registering in this study a Vacuity Index of 71.08%. This result is within the parameters reported by other authors.

Cabral et al. [36] reported that a high rate of voided stomachs may be related to a high rate of evacuation between the stomach and the intestine, such that voided stomachs would not be a true measure of feeding activity. Furthermore, using the weight of food in the entire digestive tract as a more reliable estimate of food consumption in this species is suggested [13,28].

Based on our results, we can state that the Vacuity Index is a parameter mainly linked to seasonality and on a lesser degree to sex.

The weight of the food content of the stomachs and the Mean Fullness Index in males was higher in terms of quantity. However, fewer full stomachs were found for this sex group. In addition, there is a clear interaction between autumn and summer seasons and sex. In autumn, males showed lower stomach contents and lower Mean Fullness Index with respect to females and seasons.

Based on our results, we can establish that male fish feed more in autumn. Likewise, males seem to feed more, probably due to their physiological requirements for reproduction [51] and their larger sizes. Males possess a larger stomach capacity; thus, they tend to have heavier stomach contents. Continuing with the analysis of our results, it can be mentioned that the intensity of feeding dynamics could be higher in small fish, possibly due to a higher digestion speed of small prey (a problem that should be corroborated in another investigation) as mentioned by Wootton [18]. Large fish can be satiated more quickly by the ingestion of large prey. Another explanation is that juveniles inhabit shallower and warmer waters, unlike adults that prefer deeper and colder waters, obtaining a benefit of low metabolic cost and greater longevity [44]. The same author states that temperature can have subtle effects on the food preferences of fish, citing the example of the *Ctenopheryngoden idella* carp, for which temperature influences the efficiency of stimulation of taste receptors for different amino acids. Stoner [44] describes that feeding motivation is determined by metabolic needs and the amount of food in the stomach. Temperature affects metabolic rates and gastric emptying in ectothermic fish, and thus foraging and feeding intensity are also temperature dependent. Food intake increases rapidly with temperature until it reaches a maximum and then decreases if the temperature is excessive, as proven in farmed fish [52]. Likewise, Stoner [44] states that light intensity also has an effect on feeding rates; light level is routinely used in aquaculture to influence fish activity and feeding.

Wootton [18] states that the amount of food required to satiate a fish is related to the distended state of its stomach. The rate of feed consumption depends on the rate at which stomach content is evacuated. The rate of evacuation increases with temperature and with the quality of the food, which also influences the rate of evacuation. Food of low energy content is evacuated faster than food with a high energy content; however, if the diet consists of a mixture of different prey, the rate of evacuation of each type of prey is not independent of the other items. Finally, Wootton [18] describes that one more factor that can affect the consumption rate is physiological state; in some species, feeding rate decreases or stops when the fish starts to be reproductively active.

Fish size is a relevant factor in their diet since fish vision improves with age; thus, larger individuals will have an additional advantage in the selection of larger prey [18,53].

### 4.2. Study of the Trophic Composition of the Diet

In the examination of a sample, the frequency of occurrence expresses the percentage of stomachs containing a given prey item. Frequency of occurrence thus describes the uniformity with which clusters of fish sort out their diets [3,35,53]. Still, it does not indicate the significance of the various types of food ingested.

In regard to trophic composition of *Sarda chiliensis chiliensis*’s diet, it can be conditioned by various factors, such as the number of studied individuals, the degree of prey digestion, the availability of prey [54], the feeding behavior of the predator [55] and the habitat or habitats occupied by it, such as the type of sediments in their environment [56]. Berg [8] describes the latter aspect that many organisms are related to their environment by their trophic affinity, controlled by the abundance and behavior between predator and prey. In this sense, as *Sarda chiliensis chiliensis* is a neritic pelagic species, it certainly explains why the main items are pelagic animals, mainly fish (*Engraulis ringens*) and shrimp (*Pleuroncodes monodon*) [57]. According to the remains of vertebrae found in their stomachs, these corresponded to anchoveta (*Engraulis ringens*). This situation coincides with what was previously described by Medina and Araya [58] and Sánchez de Benites et al. [59], that anchoveta was the preferred species in their diet. This is also consistent with a detailed study of the feeding habits of *Sarda chiliensis lineolata* by Pinkas et al. [37], based on a total of 1498 stomachs sampled between 1968 and 1969. The results clearly showed that northern anchoveta *Engraulis mordax* was the major component (75.9%) in the diet of *Sarda chiliensis lineolata* and that common squid (*Loligo opalescens*) was the next most important (18.0%). The rest consisted of miscellaneous fish and a few crustaceans.

It is highly relevant for fish feeding studies to consider sample size in order to describe and make diet comparisons. It is suggested to evaluate this aspect before sampling or analysis, as insufficient sample size could lead to inaccurate conclusions.

As indicated by the geometric importance graphs (Figure 7, Figure 8 and Figure 9), the hierarchy of prey items in the diet of *Sarda chiliensis chiliensis* in all three study areas reveals that *Pleuroncodes monodon* and *Engraulis ringens* are the main food items.

Within the framework of combined indexes, we can state that the RI proposed by George and Hadley [60] has some advantages over others.

First, we can observe that on a 100% percentage scale, especially with respect to the IRI suggested by Pinkas et al. [37] and modified by Hacunda [61], the significance of each prey item with respect to the stomachs that presented content is highlighted [62].

Second, with respect to the Preponderance Index (PI), only the Gravimetric Index and the Frequency of Occurrence are taken into account; in contrast, the RI involves, in addition to these two, the Numerical Index.

Third, the RI gives equal importance to the Numerical, Gravimetric and Frequency Indexes by making a simple sum of the three. The IRI, in contrast, multiplies the sum of the two previous ones by the frequency.

### 4.3. Evaluation of the Food Strategy

Regarding the evaluation of feeding strategy, Amundsen [43] postulates that feeding studies are mainly based on population data, while the feeding strategy of individual fish has hardly been studied. He further establishes that it is important to clearly distinguish between the niches of different individuals and the niche of the total population. A population with a narrow niche is necessarily composed of individuals with narrow and specialized niches, while a population with an ample niche could be composed of both individuals with ample or narrow niches, or a combination of both.

Cabral et al. [36] established that the apparent dietary specialization, or at least the extremely narrow niche breadth, contradicts the status of generalists and opportunists that has been assigned to many fish species, as is the case of *Sarda chiliensis chiliensis*.

Regarding the importance of prey in the diet of *Sarda chiliensis chiliensis*, only *Engraulis ringens* and/or *Pleuroncodes monodon* were the dominant prey in the three study areas. Given the Gravimetric Index of the other prey, we can speculate that the strategy tends to make *Sarda chiliensis chiliensis* a generalist species, as reported by Medina and Araya [58].

Wootton [18] established that the analysis of fish diets is somewhat complex, since many food items may be present, with some in small quantities. Fish sampled at the same time and in the same place could have different stomach contents.

### 4.4. Study of Trophic Relationships

Regarding the studies of trophic relationships (Shannon’s Diversity Index), Chanavayita had the highest index (1.50), which coincides with the largest individuals (45.4 cm on average). “La Capilla” had a slightly lower index (1.07) with smaller individuals (36.4 cm on average), and Pozo de Lisas with medium-sized specimens (39.9 cm on average) had a 1.35 index. These results allow us to infer that larger specimens could access a greater diversity of prey; a finding consistent with Magnuson and Heitz [63], who established that the scombrids were selective eaters, especially the larger predators that have reduced their ability to capture small prey due to a larger opening between gill plates. Among scombrids of the same size, Bonitos have the largest openings between gill plates (1.8 to 3.3 mm), with an inverse relationship between the number of gill plates and gill slit (the greater the number, the smaller the slit). In this sense, Collette and Chao [64] reported that *Sarda orientalis* has fewer gill plates than *Sarda chiliensis chiliensis*, and as shown by Magnuson and Heitz [63], the gill plate opening is larger in *Sarda orientalis*; thus, it can be expected that the diet of *Sarda chiliensis chiliensis* has a higher proportion of smaller organisms than in *Sarda orientalis*. In turn, Blasković et al. [65] described that there is selectivity in terms of food size, given that the stomach contents of large and small fish of the same species are dissimilar, partly reflected in the difference in the Shannon Index.

Furthermore, we concluded that according to the results provided by the Shannon–Wiener Index and adapting to the criteria of Berg [8], the feeding character of *Sarda chiliensis chiliensis* on the coasts of northern Chile and southern Peru would be ichthyophagous and stenophagous (having values less than 3). Said data would conclude for a not diverse diet preferably made up of *Engraulis ringens* (anchoveta), *Pleuroncodes monodon* (dwarf shrimp), other prey depending on its availability, and prey of smaller size than its counterpart in the northern hemisphere.

The obtained data show that, in the three study areas, the diet is not diverse, although in the Chanavayita area, the Shannon Index value was slightly higher than in the other two study areas, indicating that there is a greater diversity of prey offered by the environment (7 prey items). Labropoulou et al. [66] studied the feeding habits of fish off the Greek coast, finding that their diet in most of the species is composed of a narrow range of prey, as presented by *Sarda chiliensis chiliensis* in this study, which was reflected in a low Shannon Diversity Index. Thus, it comes as no surprise why they consider these species to be specialists. They qualify, however, in that the dietary breadth increased with the size of the fish, which would indicate a tendency to a more generalist feeding during the later stages of growth. These data are in agreement with ours, since the individuals studied in the present investigation are adult specimens.

Regarding the homogeneity of the diet, the three study areas present uniformity among them, producing a value between 0 and 1 consistent with the diversity, since *Sarda chiliensis chiliensis* maintains a low diversity and similar homogeneity in the three areas.

Cabral et al. [36] studied the feeding ecology of several fish species and concluded that their dietary breadth is extremely narrow, as some species feed on few types of prey, consuming almost exclusively polychaetes and bivalves. Our results were similar to those found by these authors, in terms of the amount of prey items consumed.

Thus, the evaluation of diet complexity was made using the Shannon–Wiener Diversity Index, finding no significant variations in diversity between the different study areas for seasonality and sex.

### 4.5. Statistical Analysis

We can state that even when there were differences in prey weight, due to environmental variables [67] in the different seasons, no significant differences were found in the three study areas for seasonality and sex.

No significant differences were found in the number of prey items for the different indexes in the study areas with respect to sex and summer and autumn seasons.

Analysis of the dynamics of a biome depends in part on measuring how organisms use their environment. One way to do this is to measure the niche parameters of one population in comparison to another. Since feeding is one of the main dimensions of a population’s niche, the analysis of its diet serves as a tool to identify niche characteristics [68].

The analysis of diet, diversity, and inter- and intra-specific interactions, particularly to evaluate different characteristics and to determine the robustness and trend of the observed patterns, requires the implementation of different statistical techniques. Furthermore, the selection of said techniques depends fundamentally on the objectives of the study and the level of analysis required [69].

## 5. Conclusions

Fish feed more during the autumn season.The hierarchy of prey items in the diet of *Sarda chiliensis chiliensis* in the three study areas reveals *Pleuroncodes monodon* and *Engraulis ringens* as the main food prey.On the northern coasts of Chile and the southern coasts of Peru, *Sarda chiliensis chiliensis* would be an ichthyophagous and stenophagous fish, i.e., its diet is not diverse, consisting mainly of *Engraulis ringens* (anchoveta) and *Pleuroncodes monodon* (dwarf shrimp).Regarding the importance of prey in the diet of *Sarda chiliensis chiliensis* in the three study areas, this would be a generalist species.

## Figures and Tables

**Figure 1 animals-12-00930-f001:**
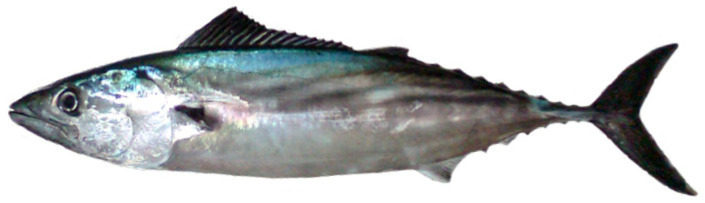
Specimen of South Pacific Bonito (*Sarda chiliensis chiliensis*).

**Figure 2 animals-12-00930-f002:**
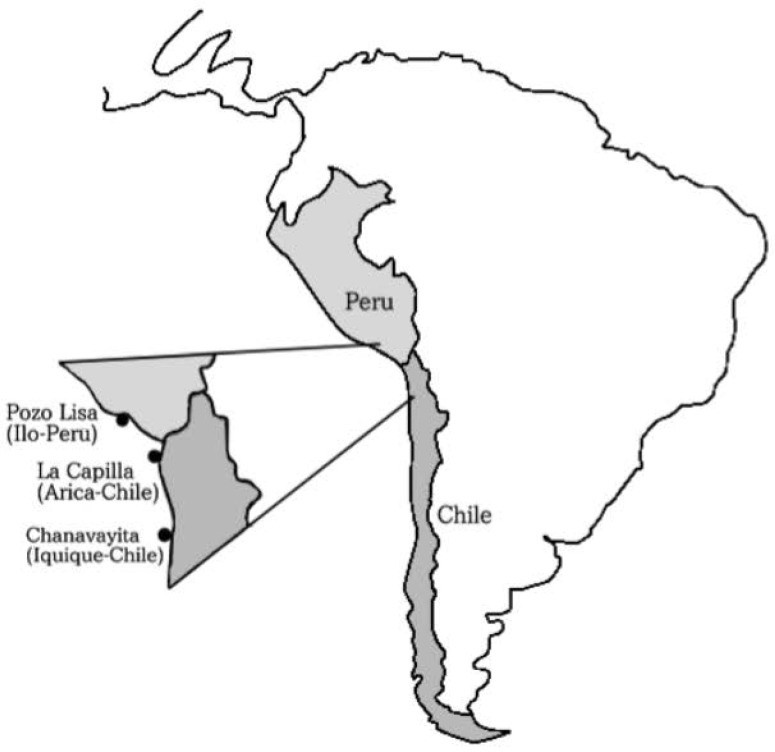
Sampling areas for specimens of *Sarda chiliensis chiliensis*: Pozo de Lisas (Ilo, Perú); Playa La Capilla (Arica, Chile); Playa Chanavayita (Iquique, Chile).

**Figure 3 animals-12-00930-f003:**
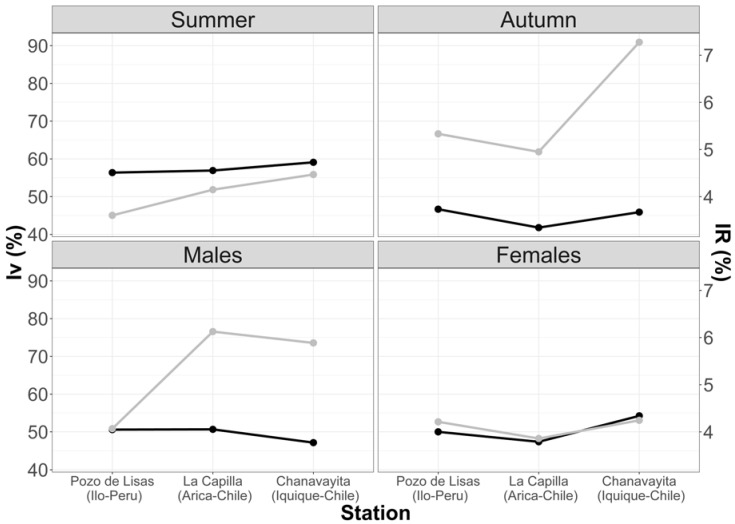
Variations in Mean Fullness and Vacuity Indexes with respect to seasons and sex. Black line (IV); grey line (IR).

**Figure 4 animals-12-00930-f004:**
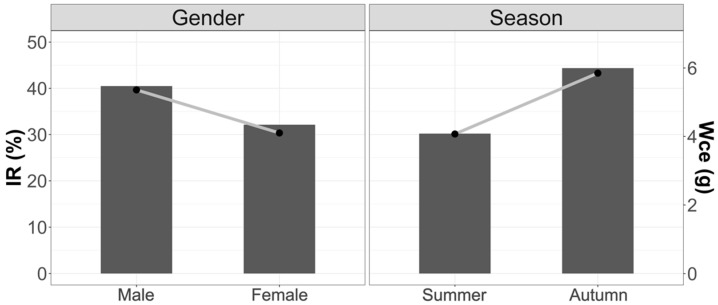
Variations in stomach weight content and Mean Fullness Index with respect to season and sex. Bars weight and line IR.

**Figure 5 animals-12-00930-f005:**
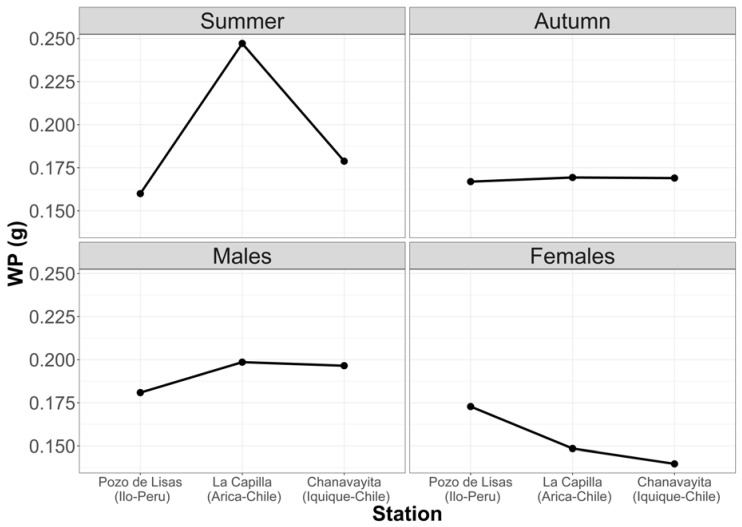
Average weight of prey with respect to seasons and sex in the three study locations.

**Figure 6 animals-12-00930-f006:**
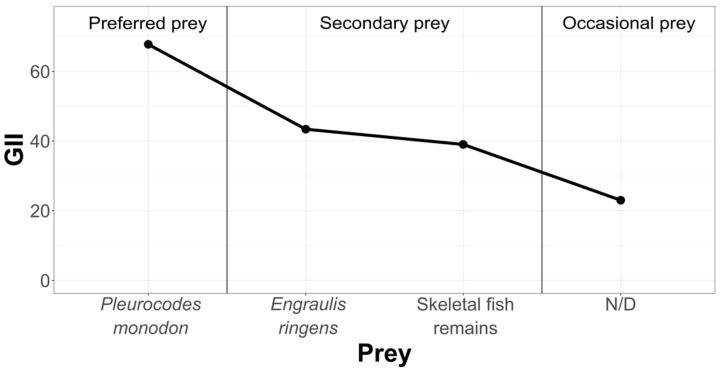
Hierarchization and classification of prey in the Pozo de Lisas study area, according to the Geometric Importance Index.

**Figure 7 animals-12-00930-f007:**
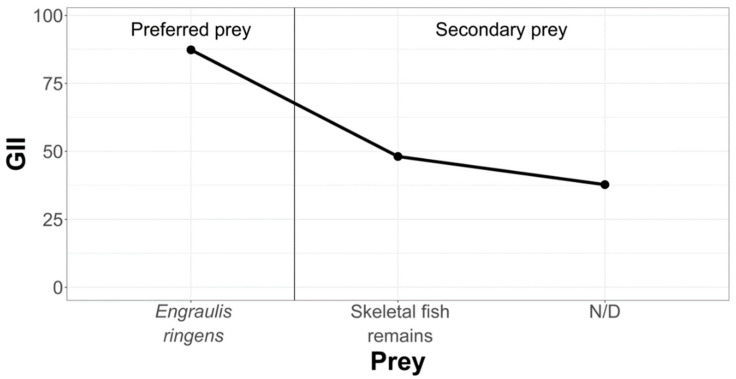
Hierarchization and classification of prey in the La Capilla study area, according to the Geometric Importance Index.

**Figure 8 animals-12-00930-f008:**
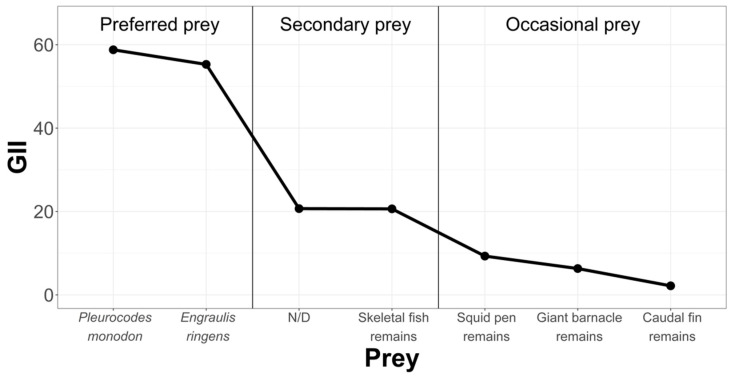
Hierarchization and classification of prey in the Chanavayita study area, according to the Geometric Importance Index.

**Figure 9 animals-12-00930-f009:**
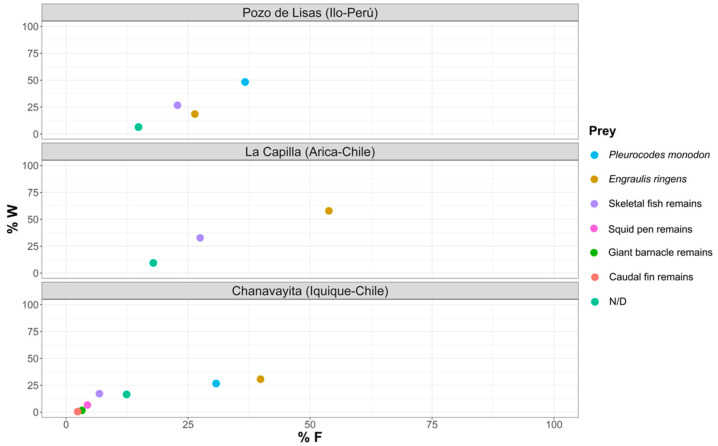
Feeding strategy for each of the study areas, according to Costello’s method [39]. Study of trophic relationships (%W represents the percentage by weight of the gravimetric method).

**Table 1 animals-12-00930-t001:** Average weight, standard length, and sex ratio of *Sarda chiliensis chiliensis* in all three sampling areas.

Capture Area	Number of Specimens Captured without Stomach Contents	Number of Specimens Captured with Stomach Contents	Total Average Weight (g)	SD (Standard Deviation)	Average Standard Length (cm)	SD (Standard Deviation)	Sex Ratio (Male/ Female)
Pozo de Lisas (Ilo, Perú)	345	145	905	173.49	39.4	1.81	99/46
La Capilla (Arica, Chile)	301	132	698	186.59	36.4	1.73	91/41
Chanavayita (Iquique, Chile)	352	129	1124	182.85	45.4	1.72	73/56

**Table 2 animals-12-00930-t002:** Frequency of occurrence (%F), Numerical (%N), Gravimetric (%W) and Relative Importance Indexes (IIR) in stomach content for *Sarda chiliensis chiliensis* in the area of Pozo de Lisas.

	Numeric Composition Index		Frequency of Occurrence Index		Gravimetric Method (g)		Relative Importance Index
Item-Prey	N	%N	F	%F	W	%W	
Skeletal fish remains	23	18.40	18	22.50	32.34	26.70	10.15
*Pleurocodes monodon*	41	32.80	29	36.25	58.56	48.34	29.41
*Engraulis ringens*	38	30.40	21	26.25	22.43	18.50	12.84
N/D	23	18.40	12	15.00	7.8	6.44	3.73
Total	125	100.00	80	100.00	121,13	100.00	

**Table 3 animals-12-00930-t003:** Frequency of occurrence (%F), Numerical (%N), Gravimetric (%W) and Relative Importance Indexes (IIR) in stomach content for *Sarda chiliensis chiliensis* in the area of La Capilla.

	Numeric Composition Index		Frequency of Occurrence Index		Gravimetric Method (g)		Relative Importance Index
Item-Prey	N	%N	F	%F	W	%W	
Skeletal fish remains	35	22.88	23	27.71	33.29	32,76	15.42
*Engraulis ringens*	60	39.22	45	54.22	58.78	57.84	52.62
N/D	58	37.91	15	18.07	9.56	9.41	8.55
Total	153	100.00	83	100.00	101.63	100.00	

**Table 4 animals-12-00930-t004:** Frequency of occurrence (%F), Numerical (%N), Gravimetric (%W) and Relative Importance Indexes (IIR) in stomach content for *Sarda chiliensis chiliensis* in the area of Chanavayita.

	Numeric Composition Index		Frequency of Occurrence Index		Gravimetric Method (g)		Relative Importance Index
Item-Prey	N	%N	F	%F	W	%W	
Skeletal fish remains	12	11.76	12	6.78	16.43	17.20	1.96
*Pleurocodes monodon*	45	44.12	55	31.07	25.45	26.64	21.99
Squid pen remains	5	4.90	8	4.52	6.40	6.70	0.52
Giant barnacle remains	6	5.88	6	3.39	1.60	1.68	0.26
Caudal fin remains	1	0.98	4	2.26	0.50	0.52	0.03
*Engraulis ringens*	26	25.49	70	39.55	29.34	30.72	22.23
N/D	7	6.86	22	12.43	15.80	16.54	2.91
Total	102	100.00	177	100.00	95.52	100.00	

**Table 5 animals-12-00930-t005:** Shannon’s dietary diversity values (H′), Shannon´s maximum diversity (Hmax), and diet homogeneity (J′).

	No. of Fish	H′	Hmax	J′
Pozo de Lisas	125	1.35	1.39	0.97
La Capilla	153	1.07	1.10	0.97
Chanavayita	102	1.50	1.95	0.77

## Data Availability

The data presented in this study are available on request from the corresponding author. The data are not publicly available for privacy reasons.

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
