# Peer review of "Feeding Habits of Sarda chiliensis chiliensis (Cuvier, 1832) in Northern Chile and Southern Perú"

_animals, 2022, doi:10.3390/ani12070930_

Round 1

Reviewer 1 Report

This is a very interesting manuscript dealing with feeding habits of Sarda chiliensis chiliensis (Cuvier, 1832) in northern Chile and southern Perú, where little information is known. However, the structure of the manuscript is not always easy to follow. For example, the statistical analysis of the data is not adequately being sufficient for the authors to present their thesis as it is not described in the section of Results. Therefore, it is not clear in the section of Results if there are any statistically significant differences between the parametres examined, while some of these differences are presented in the section of Discussion where they shouldn’t. There are also several parts of the manuscript, especially in the sections of Introduction and Discussion that are not properly structured mostly due to the language and the style that need to be improved. Additionally, the scientific information when used by the literature, most of the times is not given in a way that would help to support goals and results of the study and sometimes this information has no relation with the study. More specifically, some parts are unreasonably extended and the authors should take into account the scope of the manuscript. Moreover, the available literature is not properly used either in the sections of Introduction or Discussion. Therefore, I do not recommend publication of the manuscript unless there is an improvement of the structure and the English language in order to remove the concerns expressed.

Author Response

The answers to the remarks of proofreader 1 are expressed in the same document (article) that is attached, due to the form he used in his corrections.

Reviewer 2 Report

Comments to the Authors

The manuscript “Feeding habits of Sarda chiliensis chiliensis (Cuvier, 1832) in northern Chile and southern Perú” presents relevant information and exhaustive data on the feeding habits of an interesting species for aquaculture. It is noteworthy that the present work analyses numerous variables in a large set of data.

However, as the manuscript is, it is very difficult to interpret the results obtained. It is suggested a major revision of the different sections to improve the clarity of the manuscript.

Major and minor points are indicated to improve the manuscript. Please see below for more specific comments.

Major issues

  1. The introduction needs to be revised in order to provide more information on previous studies on this subject and the necessity of further research. In this regard, the paragraph from L96 to L100 would be better included in the introduction section. The phrase in L87-88 should me moved to the section about the study of stomach content.
  2. L93-94 are confusing. Out of curiosity, is it required a specific permit for fishing this species in Chile?
  3. The manuscript would greatly benefit if all the abbreviations of the calculated variables would be consistent throughout the text. For instance: in L163 the Mean Fullness index is abbreviated with FI, while in the formula it is IR, as well as in the graphics (results).
  4. It is recommended to provide the statistical differences found in the Results section (also in the figures and tables). The type of differences obtained for the numerous variables would be clearer.
  5. One important part to be revised is the Results section. Firstly, the manuscript would greatly benefit if the table and figure captions would be more complete and specific. It is suggested that, if the number of analysed fish was 1404, but around 1000 had an empty stomach, the results should also include these last fish, in particular, in table 1. Did the authors observe any food item in the other digestive tract part of fish with empty stomach? Moreover, if 6 categories were considered for prey items, it is confusing that giant barnacle would be considered as an extra category just in one study area.

Figure 4-6 is referring to the mean values for the analysed fish? Moreover, it would be recommended that two vertical axis graphics would have a legend to identify each variable.

Table 2-4: please include the correspondent study area.

  1. In the discussion section it would be recommended a careful revision for clarity. First, it is important to note that the results regarding the male size with the seasonality are not depicted in this study (L457-459), but it would be very interesting to show. Moreover, this study analysed the feeding dynamics regarding sex and seasonality, not the size of the fish (L460-465). L488-L491: are very confusing and relatively detached from other line of though. It is important to mention that the present study does not analyse the effect of water temperature, light intensity or metabolism in feeding dynamics. This paragraph should be rephrased.

L508-509: these results are not shown in the results section.

L519: This information should be in results section

L596-603: This important part of the study should be moved to the Results section.

L460; 488, 623: more favorable times in terms of trophic supply – for a direct approach, this phrase could be substituted by Autumn; is there any reference that provides information on the major amount of food in the Autumn season?

Minor issues

L115. Please modify the sentence. As example: Each individual was dissected in order to extract their stomachs and then sexed. Moreover, to calculate the Mean Fullness index it is required to eviscerate the fish, which should be stated in M&M section.

L132-134: It could be moved to the results section.

L187-190: This paragraph could be in the discussion section.

L283: The sentence requires further citation, and it would be better to move it to the discussion.

L417: bonito populations

L440: could the authors please specify which authors?

L455: weight of the food content of the stomachs (Mean fullness Index) – these are two different variables.

L462: is that a general occurrence in South pacific Bonito, or could the authors please provide some references?

L497: again, the weight of the prey was analysed considering sex and seasonality, not fish size.

L509: ringen

L554: for trophic relationships, 2 indexes were analysed, which one are the authors referring to?

Table 2: please rephrase the words in Spanish

Figure 10: please provide the meaning of the variable % W (it is not mentioned in M&M section).

Author Response

Attached PDF

Reviewer 3 Report

The paper provides some interesting insights with respect to the trophic ecology of South Pacific Bonito. However some parts of the manuscript should be presented in a clearer way, particularly in the discussion, which would need to be streamlined or maybe divided into sections.

line 75. please add some reference related to this decline

line 77: please indicate the subject

line 84: please change specimens with species

lines 93-95: should be moved to paragraph 2.1

lines 96-100: this part belongs to introduction (maybe between line 76 and 77?)

lines 270-274: are these differences significant?

lines 410-413: not clear, please rephrase

line 429: change full stop with comma

line 430: delete "as" after the semicolon

line 461: any studies about their physiological requirements for reproduction?

line 474: do you mean ectothermic?

line 505: the entire period is not clear, maybe replace the semicolon with a comma or rephrase

line 531-533: not clear, please rephrase.

The discussion is quite long it should be shortened and re-organized, maybe creating some sub-sections, in order to highlight the results obtained in the study with respect to previous information, and make easier for the reader to extract the main concepts.

Author Response

Attached PDF

Round 2

Reviewer 1 Report

see the comments in the manuscript

Author Response

I send two files one in PDF and the other one in Word, the first one has the language modifications

I can't upload the two files, but the word file is also sent to the reviewer two

Reviewer 2 Report

Comments to the Authors

The manuscript “Feeding habits of Sarda chiliensis chiliensis (Cuvier, 1832) in northern Chile and southern Perú” presents relevant information and exhaustive data on the feeding habits of an interesting species for aquaculture. It is noteworthy that the present work analyses numerous variables in a large set of data.

The manuscript was greatly improved regarding clarity.

Major and Minor issues

L71-74. Please divide this sentence in three sentences, for clarity.

L167…: please be consistent with the variables in the text and in the formulas: VI is not Iv and Wsc is not Wce; NC% is not CN(%); RII is not IRI ?

L262: Vacuity Index (VI)

Table1: If Table 1 is related with all the sampled fish, the results should be related with all the fish. With this it is suggested that the total weight, length and sex ratio should be related with all the 1404 fish. Other option is to specify that the table is only related with the fish with stomach contents, both in the text as in the table caption. It should be mentioned in the text that the remaining results are related only to the specimens with stomach contents (about 400 fish), if this is the case.

L280-282: The reviewer apologizes, the line that should be inserted in results should be L509 (v1): It is also worth noting that according to the remains of vertebrae found in their stomachs, these corresponded to anchoveta (Engraulis ringer).

L284: Vacuity index (VI) and Mean Fullness Index (IR). Please be consistent with abbreviations

L308: Do you mean Figure 3?

Figure 3 and figure 4: The values detailed are correspondent to mean values?.

Figure 4: Please indicate which variables correspond to bars and lines.

Table 4: please rephrase the words in Spanish

L537: Do you mean Figure 9?

L595: Please add the authors.

L608: could be a parameter

L611: this sex group

L613-614: Is this important information depicted in the results?.

L616-618: The authors tried to include some references to the sentences. Since the data on male size was not provided, it is suggested to add more references regarding the overall higher size of males and their stomach capacity.

L617: please put the references in the correct format.

L621-622: Please provide some references to this sentence.

L675: Berg[8]!Please correct.

L679-680: These results should be added to the results section (in order to be more clear in the discussion).

Author Response

There are corrections in the text
